# Neural Decoding through Multi-subject Class-conditional Hyperalignment

## Abstract

Understanding brain dynamics in multi-subject studies is challenging, as each individual exhibits unique neural patterns. Such variability complicates the identification of shared task-related dynamics without carefully accounting for meaningful individual differences. Typical analyses involve fitting subject-specific models separately and aggregating results post hoc. This approach, however, precludes the possibility of information sharing across the models. Hyperalignment methods resolve this by mapping subject-specific responses into a shared latent representational space, but typically require a secondary dataset to learn these mappings by exposing all subjects to an identical, rich and evocative stimulus, such as watching an exciting movie. These datasets are costly to collect and understandably infeasible in nonhuman studies. An alignment method for multi-subject studies that can be applied directly to the primary dataset would be of immense value. To this end, we introduce the **Mu**lti-**S**ubject **C**lass-conditional **H**yperalignmnet (**MuSCH**) model which learns aligned latent embeddings of multi-subject data by leveraging class labels available from the experimental protocol of the primary dataset itself. **MuSCH** trains subject-specific encoder networks using a novel Supervised Contrastive Learning framework which simultaneously makes same-class embeddings similar and different-class embeddings dissimilar across subjects. Using both simulation studies and a real memory experiment, we demonstrate how principled information sharing improves the performance of a downstream neural decoding task. Furthermore, by modulating signal strength in the simulated dataset, we show that classification improvements are especially pronounced in regimes with weak signals, a situation commonly encountered in neuroscience investigations. **MuSCH** obviates traditional hyperalignment's onerous prerequisite of a secondary alignment dataset, extending the promise of a single robust and generalizable model to any labeled, multi-subject dataset where subject-specific distortions prevent a joint analysis.

## 1 Introduction

A fundamental challenge in neuroscience is leveraging data across multiple subjects to improve our understanding of brain dynamics. While individual brains exhibit unique anatomical and functional characteristics, the underlying neural processes that support behavior are presumed to share common computational roles. However, translating this assumption into practical analytical frameworks remains difficult, particularly when working with heterogeneous neural recordings that vary in their spatial, temporal, and statistical properties across subjects.

This challenge is especially acute when analyzing neural data collected from different subjects in a randomized experimental setting. Differences in positioning of the recording devices, recording quality, individual differences in neural organization, and subject-specific neural encoding patterns all contribute to data heterogeneity that can obscure shared neural structure. Traditional approaches often treat each subject's data independently, potentially missing opportunities to incorporate cross-subject information that could improve analytical power and reveal common underlying mechanisms. More recent efforts to align multi-subject neural data to a common latent space prior to analysis have shown some success (Haxby et al., 2011; Chen et al., 2015; Guntupalli et al., 2016). However, these "hyperalignment" methods all require a secondary dataset collected from the same subjects used for the alignment process. This secondary dataset must be time-locked, i.e. all sub-

jects are exposed to an identical stimulus at the same time over the course of collection. Typically subject are shown an evocative movie, such as Indiana Jones, Raider of the Lost Arc (Haxby et al., 2011), or a long audio clip. Collection of this secondary data set is time-consuming. It is also not included in the experimental protocol for most existing datasets which precludes hyperalignment. Further, with non-human subjects collection of this secondary data set is not possible. It is clear that a hyperalignment method that can be applied directly to the primary experimental dataset would have immediate value.

There also exist supervised functional alignment methods based on generalized canonical correlation analysis (CCA), including MaxVar-GCCA and stimulus-informed variants such as SI-GCCA (Shu et al., 2022; Fu et al., 2017; Geirnaert et al., 2023), which learn subject-specific linear projections into a shared latent subspace using a shared stimulus as an additional view. These methods, however, similarly assume views arise in response to a common stimulus. No existing GCCA variant can accommodate data from a randomized-trial design where there is no natural one-to-one trial correspondence across subjects.

The use of transformers have recently demonstrated an impressive ability to distill shared patterns in multi-subject neural spike data to accomplish a prespecified downstream task. POYO decodes a single mode of subject behavior from neural spike data (Azabou et al., 2023); POYO+ decodes several modes simultaneously (Azabou et al., 2025); NEDS works bidirectionally, decoding spike data from behavior and vice-versa across a wide range of experimental tasks (Zhang et al., 2025). While the performance of these models on a wide range of heterogeneous data sets from multiple labs and experiments demonstrate an impressive ability to distill common patterns of activity, these common patterns are not accessible. These models do not yield interpretable common latent representations of neural responses to stimuli which can then be used for any downstream task. These latent representations are valuable in applied neuroscience because they can reveal shared relationships that are not readily discernible in individual subjects.

**Contributions.** Our contributions are twofold, encompassing both methodological advancements and practical applications. First, we propose a novel alignment method that maps heterogeneous multi-subject data into a shared latent representational space, preserving class structure and enhancing class separability for downstream tasks. This approach combines Supervised Contrastive Learning (SupCon) with independent subject-specific encoders. Second, we demonstrate that this framework can improve the alignment of latent embeddings from neural data across multiple subjects, thereby enhancing the performance of downstream neural decoding. Specifically, we show that our method improves decoding accuracy on both controlled simulations and real hippocampal data from rats performing a non-spatial sequential memory task. The improvements are especially pronounced in low-signal regimes, a common challenge in neuroscience. While SupCon has proven effective in other domains, its application to neural data for noise filtering and the identification of shared brain dynamics is novel. By eliminating the need for extensive secondary training datasets required by existing hyperalignment methods, our approach significantly broadens the range of multi-subject neural datasets that can be used for latent representational alignment.

## 2 RELATED WORK

**Hyperalignment.** All modalities of neural data present multiple sources of variability including scanner noise, physiological artifacts, and subject-specific functional organization. Within fMRI studies, spatial normalization to standard templates (Talairach & Tournoux, 1988; Evans et al., 1993) mitigates anatomical misalignment but not functional variability. Hyperalignment addresses this by learning subject-specific linear maps that rotate each subject's responses into a common representational space (Haxby et al., 2011). The Shared Response Model (Chen et al., 2015) generalizes hyperalignment to lower-rank latent representations, and later variants improve spatial localization (Guntupalli et al., 2016) and flexibility via autoencoder architectures (Huang et al., 2022). A significant limitation of all these methods, however, are their reliance on a secondary training dataset where all subjects are exposed to an identical, rich and evocative stimulus, such as watching an exciting movie, providing weak labels via temporal correspondence across subjects. Such data are costly and time-consuming to collect and are understandably infeasible in nonhuman studies. A direct approach to aligning heterogeneous multi-subject data would expand the range of datasets amenable to alignment, yet is largely unexplored.

**Supervised Contrastive Learning.** The challenge of learning meaningful representations from heterogeneous data extends beyond neuroimaging to the broader machine learning community. Supervised Contrastive Learning (SupCon) (Khosla et al., 2020) provides a particularly relevant framework that shares conceptual similarities with hyperalignment while addressing the alignment problem differently. Like hyperalignment, SupCon learns mappings to a latent space while enforcing similarity constraints, but whereas hyperalignment treats time as the implicit class label (aligning neural activity at timestamp $t$ across subjects), SupCon uses explicit categorical labels to cluster samples from the same class together regardless of other sources of variation. Despite broad success in shaping discriminative latent spaces, SupCon's value in application to multi-subject alignment has not, to our knowledge, been established.

**Multimodal VAEs.** Multimodal variational autoencoders learn a shared latent space for co-registered observations from different sources, balancing per-modality reconstruction with a prior that encourages their latent embeddings to coincide. Alignment is driven by the choice of prior. The MMVM-VAE uses a data-dependent mixture-of-experts prior to "soft-share" information across views of the same physiological process (Erlacher et al., 2025). The MMVM-VAE was also shown by the original authors to align responses from multiple subjects exposed to the same stimulus by treating subjects as distinct modalities (Sutter et al., 2024), demonstrating how, in principle, it can achieve the alignment we seek. However, the mixture-of-experts prior produces only soft, partial within-class alignment and does not explicitly separate different classes. Consequently, its latent embeddings may be suboptimal for downstream decoding.

## 3 METHODS

We address the lack of an alignment method that can be directly applied to the primary multi-subject data by applying the objective of Supervised Contrastive Learning to a generalized form of the hyperalignment model. Our proposed method, which we call "**Mu**lti-**S**ubject **C**lass-conditional **H**yperalignment (**MuSCH**)" learns transformations that map the primary dataset to a common latent space using class labels rather than learning these transformations on a secondary dataset according to co-occurrence in time. We discuss the details below.

**Multi-subject Class-conditional Hyperalignment.** We first introduce a generalized hyperalignment model that serves as a common framework for classical hyperalignment, the Shared Response Model (SRM), and our proposed **MuSCH** method. Consider neural data from $N$ subjects exposed to $K$ stimulus classes. Let $y_{ij}$ be the class corresponding to trial $j \in \{1, ..., n_i\}$ of subject $i \in \{1, ..., N\}$ and $x_{ij} \in \mathbb{R}^{P_i}$ denote the observed neural response, where $n_i$ is the number of samples taken from subject $i$ and $P_i$ is the number of features (e.g., voxels or neurons). Samples from subject $i$ are commonly stacked as rows in $X_i \in \mathbb{R}^{n_i \times P_i}$. In fMRI, the rows of $X_i$ are consecutive scans and the columns are voxels within a region of interest; in spike-train data (our setting), $X_i$ contains binned firing rates, with rows as trials and columns as neurons. We model each observation as a noisy mapping into a shared, class-structured latent space. For subject $i$ and trial $j$, let $z_{ij} = f_i(X_{ij}) \in \mathbb{R}^D$ denote the latent representation produced by a subject-specific encoder $f_i : \mathbb{R}^{P_i} \to \mathbb{R}^D$.

We obtain a generalized form of the hyperalignment model by defining

$$f_i(x_{ij}) = S\delta_{ij} + \epsilon_{ij} \tag{1}$$

where the columns of $S \in \mathbb{R}^{D \times K}$ are some $D$-dimensional latent representations of the $K$ experimental classes (its $k$-th column is the representation $S_k$), $\delta_{ij} \in \{0,1\}^K$ is the one-hot encoding of class label $y_{ij}$, and $\epsilon_{ij}$ is zero-mean noise. We refer to Equation 1 as the *generalized hyperalignment model*: it specifies a shared latent representation $S$ and subject-specific transformations $f_i$ that map observed data $x_{ij}$ into this shared space in a class-conditional manner. When all subjects are exposed to the same time-locked continuous stimulus, such as watching a movie, and the data collected are fMRI scans let $T$ denote the number of time points (TRs) in the shared stimulus. In this case we have $K = T$, $y_{ij} = j$ for all $i$, and $n_i = T$ for all $i$. If we restrict $f_i$ to be linear with $f_i(x) = W_i^\top x$ and $W_i^\top W_i = I_D$, then Equation 1 reduces to classical hyperalignment of Haxby et al. (2011). Under the same linear constraint with $D < P_i$, Equation 1 is algebraically equivalent to the Shared Response Model (SRM) of Chen et al. (2015).

The generalized form of the hyperalignment model in Equation 1 suggests other ways to achieve alignment of multi-subject data through the choice of form for the subject-specific transformations $f_i$, the choice of label represented by $\delta_{ij}$ and the choice of training objective (hyperalignment is implemented using least-squares). Our **MuSCH** model can be viewed as another instantiation of Equation 1 where $f_i$ are neural networks, $\delta_{ij}$ are class labels from the primary data set, and the training objective is the SupCon loss function, details of which we discuss next. An overview of the MuSCH framework and the baseline methods is shown in Figure 1A.

**Encouraging Latent Intra-class Similarity and Inter-class Dissimilarity.** To align embeddings $z_{ij} = f_i(x_{ij})$ at the class level while aiding downstream decoding, we use the SupCon loss function (Khosla et al., 2020). Let $A = \{(i,j)\}$ be the set of indices for the observations in a training mini-batch, where $i$ is the subject index and $j$ is the trial index. For an anchor $(i', j') \in A$ define the set of all indices in the training mini-batch having the same class label as $P(i', j') = \{(i,j) \in A \setminus (i',j') : \delta_{ij} = \delta_{i'j'}\}$. These are known as "positives" in the contrastive learning literature. The per-anchor SupCon loss is defined as

$$\mathcal{L}(i', j') = -\frac{1}{|P(i', j')|} \sum_{p \in P(i', j')} \log \frac{\exp\left(z_{i'j'} \cdot z_p / \tau\right)}{\sum_{a \in A} \exp\left(z_{i'j'} \cdot \hat{z}_a / \tau\right)}, \tag{2}$$

with scalar temperature $\tau > 0$. The batch loss is the sum of the per-anchor losses.

This objective pulls together embeddings that share the same class label (positives) while simultaneously pushing apart embeddings from different classes, within and across subjects. By constructing $A$ to include trials from multiple subjects and classes, independent encoders $f_i$ are trained to produce subject-agnostic, class-consistent representations. In contrast, classical hyperalignment uses only positive pairs defined by co-occurrence in time and does not explicitly repel embeddings from different classes; SupCon provides both intra-class attraction and inter-class separation, which is advantageous for downstream classification. We provide justification that this choice of loss function encourages network configurations in accordance with Equation 1 in Appendix A.1.

## 4 IMPLEMENTATION DETAILS

The methods we consider in this study are illustrated in Figure 1. We evaluate the performance of a downstream decoder under four different setups where only the source (single-subject vs. multi-subject) and form (raw vs. aligned) of the training data differ.

### 4.1 THE MODELS

**Raw Data, Single Subject (Raw-Single)** The decoder is trained directly on the target subject's raw features using only that subject's training split. No alignment via SupCon is used. This reflects the standard within-subject baseline common in neuroscience, where models are fit independently per subject and performance is subsequently summarized across subjects.

**Raw Data, All Subjects (Raw-Pooled)** The decoder is trained directly on the union of raw training data from the target subject and all data from all other subjects without any alignment. Evaluation is on the target subject's heldout test data. This "naive pooling" approach represents an ablation of **MuSCH** that removes alignment effects. It tests whether more data alone improves accuracy despite cross-subject non-exchangeability.

The remaining two setups use a two-step training process. Subject-specific encoders are first trained using SupCon loss to produce subject-agnostic latent representations. The decoder is then trained on these representations using cross-entropy loss. We describe those setups next.

**Aligned Data, Single Subject (Aligned-Single)** We train a subject-specific encoder $f_{target}$ on the target subject's training split using SupCon loss. This has the effect of making the latent representations of observations from the same class similar, aligning them. The downstream classifier is then trained on the latent representations of only the target subjects's training data. This can be viewed as an ablation of **MuSCH** because it isolates the effect of representation learning (intra-class compactness, inter-class separation) without leveraging cross-subject information.

**Aligned Data, All Subjects (MuSCH)** This is our proposed model. We train one encoder per subject ($f_i$) jointly on pooled multi-subject batches with SupCon loss, where positives are any trials sharing the same odor label across subjects. After alignment, we train the classifier on the pooled latent representations and evaluate on the target subject's holdout (encoded by $f_{target}$). This setup tests whether cross-subject alignment improves generalization to the target subject beyond what naive pooling (**Raw-Pooled**) or within-subject latents (**Aligned-Single**) can do individually.

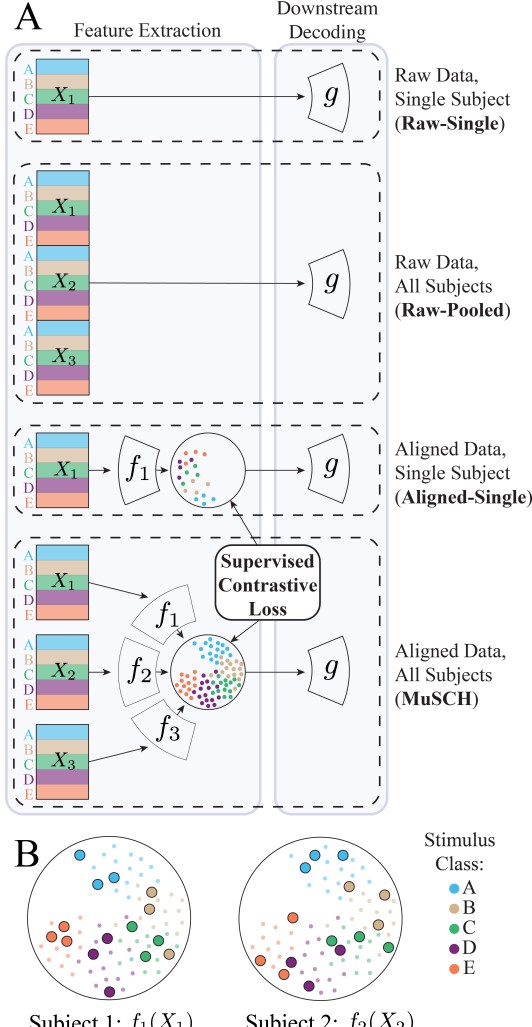

### 4.2 ENCODER DETAILS

**Architecture.** Each subject $i$ has a shallow encoder $f_i$ that maps padded neural features to a unit-norm latent vector. These encoders consist of a drop-out layer, to avoid overfitting, followed by a dense hidden layer with linear activation. The output is normalized to the unit hypersphere, as was done by Khosla et al. (2020). Normalization prepares embeddings for cosine-similarity-based SupCon. The input dimension for each encoder is the number of features in the raw data, described below, while the dimension of the latent output is $D = 15$.[1] We ensure that the target subject's encoder $f_{target}$ is initialized with the same weights in both the **Aligned-Single** setup and the **MuSCH** setup to avoid differences in initial weights affecting the quality of the learned representations.

**Training.** Encoders optimize their own portion of the SupCon loss in mini-batches. The only difference across setups is how batches are composed and routed through subject-specific encoders. In the **Aligned-Single** setup each mini-batch contains only trials from the target subject. We apply the target encoder $f_{target}$ to all inputs, L2-normalize the embeddings, and compute the SupCon loss using same-label trials as positives. Gradients update $f_{target}$ only. In the **MuSCH** setup mini-batches come from pooled data across all rats. For each trial $x_{ij}$ a switching mechanism routes $x_{ij}$ to the corresponding encoder $f_i$. The resulting unit-norm embeddings $z_{ij} = f_i(x_{ij})$ are concatenated back into batch order, and the SupCon loss is computed for each sample in the pooled batch, treating trials with the same odor label, regardless of subject, as positives. Gradients from $z_{ij}$ are back-propagated only through their originating encoder $f_i$; no cross-subject parameters are updated. Additional training details are provided in Appendix A.2. After SupCon training, encoders are frozen and the downstream decoder is trained on the resulting latent embeddings, as described below.

Figure 1: **A.** Schematic summary of the different inputs we consider to the downstream classifier $g$. **Raw-Single** trains $g$ on the target subject's raw training data alone. **Raw-Pooled** trains $g$ on the target subject's raw data as well as the raw data from the other subjects. **Aligned-Single** trains $g$ on the latent representations of the target subject's training data after it has been aligned using the SupCon loss. **MuSCH** trains $g$ on the latent representations from all subject's training data aligned using the SupCon loss. **B.** An illustration of the latent embeddings from **MuSCH** with subject 1's embeddings emphasized on the left and subject 2's embeddings emphasized on the right.

---

[1]We tried a range of latent dimensions and found that larger values of $D$ led to unused capacity (low-variance dimensions and near-zero decoder weights) while smaller values inhibited class separability; a latent dimension of 15 struck a good balance.

## 4.3 DECODER DETAILS

The decoder ($g$ in Figure 1) is a compact feed-forward network composed of a single fully connected hidden layer with sigmoid activation, followed by a softmax output over odor classes to produce class probabilities. We optimize with categorical cross-entropy, which we also use as the performance metric to compare methods. The network architecture is identical across all models; only the input dimensionality differs. **Raw-Single** and **Raw-Pooled** are trained on raw data $x_{ij}$ so the input dimension is the number of feature in the raw data. **Aligned-Single** and **MuSCH** are trained on the latent embeddings so the input dimension is $D = 15$. Justification for our choice of decoding model architecture and an exploration of a simpler linear model can be found in Appendix A.4.

## 5 EXPERIMENTS AND RESULTS

In this section, we first describe the datasets and experimental design, then the preprocessing pipeline and model architecture, followed by evaluation metrics and results.

### 5.1 DATASET

We evaluate **MuSCH** on a real hippocampal sequence-memory dataset and a simulated dataset designed to mirror the real dataset's key characteristics. We first describe the real dataset (task, subjects, recordings) and then the simulated dataset. Preprocessing and augmentations are detailed in the next subsection.

Our study uses a well-established non-spatial sequence-memory task collected from five male Long-Evans rats, with strong behavioral correspondence to human performance (Allen et al., 2016; Shahbaba et al., 2022). The rats discriminated "in sequence" (InSeq) versus "out of sequence" (OutSeq) odor presentations within repeated five-item sequences (e.g., ABCDE) delivered through a single odor port on a linear track. On each trial, rats made a nosepoke to sample an odor, holding for 1.2 s for InSeq or withdrawing early for OutSeq (including repeat and skip probe trials) to obtain water rewards (Figure 2.a). After training to criterion ($\geq 80\%$ accuracy), animals were implanted with 20-tetrode microdrives targeting the CA1 pyramidal cell layer of the dorsal hippocampus. Neural recordings comprised single-unit spiking activity and local field potentials (LFPs) as rats performed the task (Figure 2.b), yielding a rich dataset for studying the temporal organization of memory and hippocampal sequence processing[2].

Our downstream task is to decode odor from a single rat's spiking activity. It is important to note that decoding non-spatial information from hippocampal neurons is considerably harder than decoding movement information from motor cortex neurons, as was done with the transformer-based model POYO mentioned in the introduction. The hippocampus is several synapses (connections) away from the external world and thus represents highly abstracted and high-dimensional information, whereas there is a very close relationship between motor cortex activity and movement. In addition, hippocampal activity is particularly sparse and variable relative to motor cortex activity. We investigate whether decoding is made easier if we consider other rats' data when training the decoder. However, trials are not exchangeable across rats: the number of simultaneously recorded neurons differs (so feature dimensions $P_i$ vary), and anatomical variability plus surgical placement lead to nonhomologous units recording slightly different dorsal CA1 subregions. Consequently, before training the decoder we align each subject's spike data into a shared latent space using **MuSCH**.

We complement the real dataset with a simulated dataset designed to mirror its key characteristics while accommodating controlled cross-subject variability. Details about this simulated dataset are described below and the code used to generate it is provided in the supplementary material.

### 5.2 PREPROCESSING

Following Shahbaba et al. (2022), we focus on the 250–500 ms window after odor onset, the period during which odor identity is most strongly represented in CA1 activity. For subject $i$, trial $j$, and neuron $n$, we compute spike counts $c_{ijn}$ in this 250 ms window and convert to rates $x_{ijn} =$

---

[2]The dataset can be downloaded here: https://datadryad.org/dataset/doi:10.7280/D14X30

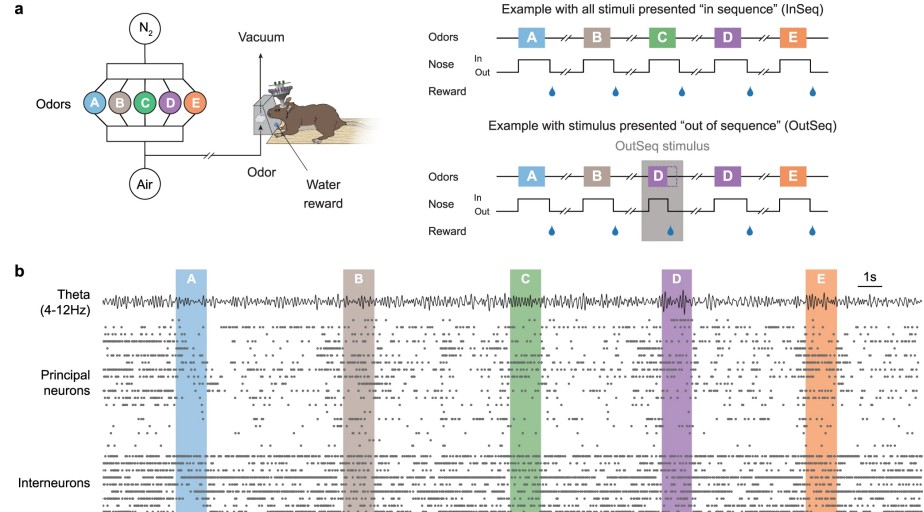

Figure 2: Odor sequence task and recording setup. Rats sample odors presented in 5-item sequences (ABCDE). InSeq trials require a 1.2 s hold; OutSeq trials require early withdrawal. Neural activity is recorded from dorsal CA1 via 20-tetrode microdrives. Image adapted from Allen et al. (2016).

$c_{ijn}/0.25s$. We include only trials in which the rat correctly reported InSeq, to focus the analysis on neural responses that truly reflect odor identity. Each observation corresponds to a single odor presentation; features are the per-neuron rates $x_{ijn}$, and labels $y_{ij}$ are odors A–E.

The number of recorded neurons varies by subject ($P_i \in [50, 104]$). While this is not conceptually a problem, our implementation using TensorFlow required model inputs to be of the same dimension across subjects. To present a fixed input size to the subject-specific encoders, we zero-pad each subject's feature vectors to $P_{max} = 104$.

## 5.3 EXPERIMENTS

We evaluate whether decoding odor identity from a target subject's neural data can be improved by incorporating data from other subjects, with particular attention to the effect of latent feature alignment prior to training the decoder. We first study a controlled simulated setting, then apply the same protocol to the real dataset. Unless noted, all preprocessing, batching, and hyperparameters are identical across setups.

**Simulated Setting.** We simulate spike data from $N = 5$ subjects (rats) and $K = 5$ classes (odors). Each subject has $P_i = 25$ neurons that all initially encode odor identity via class-specific firing rates; these firing rates differ across subjects. We iteratively treat each subject as the target subject, whose held out test data we seek to decode. For the four non-target subjects, we generate 20 trials per odor; for the target subject, we generate 80 trials per odor, reserving random samples of 60 per odor for testing and using the remaining 20 per odor for training. To achieve complex, idiosyncratic topologies, each subject's simulated data are passed through a fixed, randomly initialized nonlinear mapping $h_i$, yielding observed features $x_{ij} = h_i(\tilde{x}_{ij})$. This construction preserves class structure while creating alignment challenges analogous to those in dorsal CA1.

To assess performance under varying signal strength, we systematically destroy odor information in a controlled fraction of neurons. For each subject, we select a proportion $p \in \{0, 0.2, 0.4, 0.6, 0.8, 0.9\}$ of columns in $X_i$ and independently permute their entries across trials within that subject. This preserves marginal statistics but removes the neurons' association with class labels. For each $p$, we train the four models and evaluate on the target subject's held-out test set, reporting categorical cross-entropy. We repeat the entire procedure 60 times, randomly sampling a different subset of 60 observations per odor from the target rat's data, to obtain performance distributions.

**Real Data Setting.** For the real dataset, we treat each rat as the target subject in turn. For each target rat, we construct a stratified sample of 30% of observations from each odor class and reserve it for testing. The remaining data for the target rat as well as all data from non-target rats are used to train each of the four models and report categorical cross-entropy on the held out test set. We repeat this procedure 60 times per target subject with new random test splits and summarize performance by the distribution of resulting categorical cross-entropy calculated from the test set. We also include decoding performance using the latent embeddings obtained by the MMVM-VAE model (Sutter et al., 2024) for comparison.

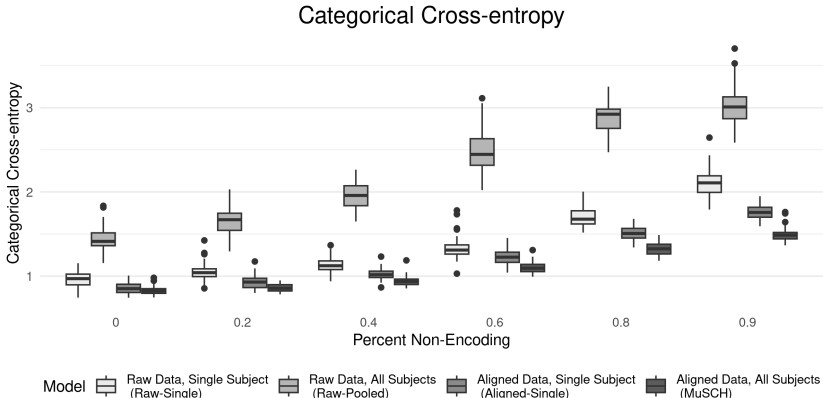

Figure 3: Simulation Results. Comparison of downstream decoding performance as measured by categorical cross-entropy across the four methods described in Section 4. "Percent Non-encoding" refers to the percent of features for each subject whose values were permuted to weaken the signal present in the data.

## 5.4 RESULTS

**Simulated Setting.** Figure 3 shows the distribution of categorical cross-entropy (cce) on the target subject's held-out test set as a function of $p$, the fraction of non-encoding neurons. As expected, cross-entropy increases with $p$ for all methods. Relative performance is consistent across seeds: **Aligned-Single** improves over the raw within-subject baseline **Raw-Single**. **MuSCH** yields the lowest cross-entropy, with the largest gains in lower signal-to-noise regimes (higher $p$). This pattern supports the intuition that when the target subject's neural response exhibits a strong signal (low $p$), there is limited headroom for additional cross-subject information; as signal weakens, SupCon-based alignment provides beneficial regularization (**Aligned-Single**) and cross-subject positives further enhance class consistency (**MuSCH**). In contrast, naive pooling without alignment (**Raw-Pooled**) performs substantially worse, indicating that cross-subject non-exchangeability harms the classifier when representations are not aligned. These results are provided in tabular form in Appendix Table 2.

**Real Data Analysis.** Table 1 summarizes the cce of the decoder according to the different setups described in Section 4 over 60 different train/test splits (these results are displayed graphically in Appendix A.4). We also include the cce when the decoder is trained on the latent embeddings learned by MMVM-VAE for comparison. The cce is split by odor to illustrate some important patterns consistent across the rats. First, odor A is the easiest to decode across all rats, likely owing to its position at the start of each sequence (highest familiarity, no interference from previous odors). For this class, **Aligned-Single** and **MuSCH** offer little or no improvement over baseline **Raw-Single**, reflecting the limited headroom for improvement when the signal is strong. Second, for harder odors (notably D and E), where per-trial signal is weaker, **Aligned-Single** reduces cross-entropy relative to **Raw-Single**, and **MuSCH** yields further reductions in many cases by leveraging cross-subject positives; these effects are seen more easily in per-class rows than in the total row, which can be dominated by the decoding performance on easier classes. The applied results mirror the simulation results: single-subject alignment acts as a beneficial regularizer when signal is low, and cross-subject alignment often provides additional improvement. Training the decoder on the latent embedding ob-

| Rat Name | Odor | Raw-Single | Raw-Pooled | MMVM-VAE | Aligned-Single | MuSCH |
|---|---|---|---|---|---|---|
| | A | $0.640 \pm 0.292$ | $3.065 \pm 1.072$ | $1.596 \pm 0.789$ | $0.599 \pm 0.299$ | $\mathbf{0.576 \pm 0.221}$ |
| | B | $2.043 \pm 0.689$ | $2.975 \pm 1.155$ | $1.536 \pm 0.537$ | $1.348 \pm 0.332$ | $\mathbf{1.262 \pm 0.341}$ |
| | C | $3.013 \pm 0.939$ | $3.828 \pm 1.576$ | $1.953 \pm 0.575$ | $\mathbf{1.485 \pm 0.452}$ | $1.707 \pm 0.510$ |
| Barat | D | $2.825 \pm 0.879$ | $5.834 \pm 1.996$ | $1.832 \pm 0.625$ | $\mathbf{1.747 \pm 0.419}$ | $1.780 \pm 0.476$ |
| | E | $1.947 \pm 0.883$ | $3.114 \pm 1.354$ | $1.723 \pm 0.854$ | $\mathbf{1.267 \pm 0.436}$ | $1.458 \pm 0.494$ |
| | Total | $1.862 \pm 0.285$ | $3.613 \pm 0.658$ | $1.694 \pm 0.343$ | $\mathbf{1.184 \pm 0.149}$ | $1.224 \pm 0.157$ |
| | A | $0.222 \pm 0.134$ | $1.245 \pm 0.606$ | $0.564 \pm 0.311$ | $0.248 \pm 0.128$ | $\mathbf{0.213 \pm 0.118}$ |
| | B | $1.349 \pm 0.593$ | $2.240 \pm 0.951$ | $0.971 \pm 0.257$ | $1.041 \pm 0.316$ | $\mathbf{0.802 \pm 0.259}$ |
| | C | $2.468 \pm 0.723$ | $3.700 \pm 1.413$ | $1.825 \pm 0.752$ | $\mathbf{1.508 \pm 0.343}$ | $1.590 \pm 0.439$ |
| Buchanan | D | $1.870 \pm 0.533$ | $3.796 \pm 1.246$ | $1.514 \pm 0.386$ | $1.234 \pm 0.285$ | $\mathbf{1.186 \pm 0.277}$ |
| | E | $1.155 \pm 0.530$ | $2.334 \pm 1.206$ | $1.005 \pm 0.5$ | $\mathbf{0.794 \pm 0.248}$ | $0.917 \pm 0.355$ |
| | Total | $1.177 \pm 0.182$ | $2.408 \pm 0.402$ | $1.051 \pm 0.176$ | $0.830 \pm 0.113$ | $\mathbf{0.789 \pm 0.103}$ |
| | A | $1.411 \pm 0.526$ | $3.106 \pm 1.069$ | $1.758 \pm 0.713$ | $1.206 \pm 0.500$ | $\mathbf{1.167 \pm 0.421}$ |
| | B | $1.450 \pm 0.443$ | $3.315 \pm 1.181$ | $1.37 \pm 0.438$ | $1.121 \pm 0.266$ | $\mathbf{1.090 \pm 0.299}$ |
| | C | $2.284 \pm 0.657$ | $4.883 \pm 1.405$ | $2.139 \pm 0.537$ | $1.801 \pm 0.380$ | $\mathbf{1.522 \pm 0.277}$ |
| Mitt | D | $1.729 \pm 0.671$ | $2.783 \pm 1.251$ | $2.144 \pm 0.856$ | $\mathbf{1.301 \pm 0.356}$ | $1.508 \pm 0.520$ |
| | E | $2.404 \pm 0.968$ | $4.096 \pm 2.033$ | $2.018 \pm 0.839$ | $\mathbf{1.527 \pm 0.573}$ | $1.735 \pm 0.692$ |
| | Total | $1.744 \pm 0.254$ | $3.566 \pm 0.535$ | $1.826 \pm 0.351$ | $1.353 \pm 0.174$ | $\mathbf{1.329 \pm 0.157}$ |
| | A | $0.318 \pm 0.238$ | $1.691 \pm 0.858$ | $0.55 \pm 0.294$ | $0.255 \pm 0.192$ | $\mathbf{0.235 \pm 0.134}$ |
| | B | $2.109 \pm 0.652$ | $3.433 \pm 0.910$ | $2.024 \pm 0.49$ | $1.598 \pm 0.323$ | $\mathbf{1.401 \pm 0.244}$ |
| | C | $2.208 \pm 0.741$ | $3.472 \pm 1.236$ | $1.461 \pm 0.375$ | $\mathbf{1.357 \pm 0.339}$ | $1.400 \pm 0.273$ |
| Stella | D | $2.711 \pm 0.887$ | $4.102 \pm 1.201$ | $2.096 \pm 0.641$ | $1.832 \pm 0.412$ | $\mathbf{1.715 \pm 0.393}$ |
| | E | $2.463 \pm 0.809$ | $4.207 \pm 1.326$ | $2.214 \pm 0.626$ | $1.936 \pm 0.456$ | $\mathbf{1.658 \pm 0.311}$ |
| | Total | $1.743 \pm 0.293$ | $3.138 \pm 0.530$ | $1.528 \pm 0.197$ | $1.244 \pm 0.134$ | $\mathbf{1.143 \pm 0.109}$ |
| | A | $0.120 \pm 0.072$ | $1.287 \pm 0.632$ | $0.395 \pm 0.426$ | $\mathbf{0.114 \pm 0.091}$ | $0.222 \pm 0.115$ |
| | B | $0.700 \pm 0.386$ | $2.107 \pm 0.970$ | $1.024 \pm 0.394$ | $0.620 \pm 0.284$ | $\mathbf{0.606 \pm 0.334}$ |
| | C | $2.260 \pm 0.772$ | $4.301 \pm 1.401$ | $1.928 \pm 0.486$ | $1.612 \pm 0.334$ | $\mathbf{1.480 \pm 0.400}$ |
| SuperChris | D | $2.237 \pm 0.946$ | $4.600 \pm 1.461$ | $1.728 \pm 0.485$ | $1.464 \pm 0.419$ | $\mathbf{1.433 \pm 0.519}$ |
| | E | $2.951 \pm 0.877$ | $4.829 \pm 1.865$ | $2.763 \pm 1.199$ | $2.093 \pm 0.657$ | $\mathbf{1.839 \pm 0.464}$ |
| | Total | $1.381 \pm 0.228$ | $3.062 \pm 0.440$ | $1.358 \pm 0.215$ | $0.997 \pm 0.124$ | $\mathbf{0.961 \pm 0.137}$ |

Table 1: Applied Results. Comparison of downstream decoding performance as measured by categorical cross-entropy (mean $\pm$ SD) across the four methods described in Section 4 and also MMVM-VAE.

tained by **MMVM-VAE** often yields better performance over training directly on the raw features of the target subject, which illustrates some degree of information sharing across subjects. However, that performance is worse than both **Aligned-Single** and **MuSCH**. As alluded to in Section 2, **MMVM-VAE** may yield suboptimal latent embeddings for downstream decoding because it does not explicitly separate the embeddings of different classes. Finally, we note that naive pooling without alignment (**Raw-Pooled**) performs substantially worse across rats and classes, underscoring the non-exchangeability of raw features across subjects and the necessity of representation alignment prior to classification.

We observe that multi-subject alignment does not yield uniform gains across all rat–class combinations. In some cases **MuSCH** improves cce, whereas in others the single-subject model performs slightly better. This heterogeneity is consistent with substantial inter-subject variability in odor representations; forcing all rats to share the same class-conditional prototypes can help when their encoding is similar but can slightly harm performance when subject-specific features are particularly informative for decoding. Importantly, across all rats and odors the performance of **MuSCH** is comparable to that of **Aligned-Single** (see mean and standard deviation in Table 1), indicating that multi-subject alignment does not systematically degrade downstream decoding while enabling a shared latent space that can be used for cross-subject analyses. We emphasize that we do not claim that **MuSCH** always yields strictly better decoding than any single-subject method, but rather that it provides a flexible generalization to classic hyperalignment methods that does not rely on a costly secondary training dataset.

An illustration of the clustering behavior induced by the Supervised Contrastive Loss function is shown in Figure 8 in the Appendix.

## 6 DISCUSSION

This study makes two important contributions to the area of supervised representational learning on multi-subject datasets. First, our within-subject alignment of latent representations of neural spike data represents the first such application of Supervised Contrastive Learning. Our results from the **Aligned-Single** model demonstrate how the SupCon loss function applied to a single subject's data can improve performance of a downstream decoder. This improvement is especially pronounced when the neural features weakly encode class information. These findings indicate that the SupCon loss serves to filter out noise across observations from the same class and subject. Our second contribution is a Multi-subject Class-conditional Hyperalignment (**MuSCH**) model which greatly expands the scope of datasets amenable to simultaneous, multi-subject analysis by removing the onerous burden of collecting a secondary training dataset required by existing hyperalignment methods. The potential application of **MuSCH** extends far beyond hippocampal spike data to any labeled, multi-subject dataset where subject-specific distortions prevent a joint analysis.

**Limitations.** A key limitation of our approach is reduced interpretability with respect to anatomical topography. Although all recordings are from dorsal CA1, neurons are sampled from different septotemporal (longitudinal) positions and subregions, which are known to support partially distinct computations. Our alignment ignores tetrode location and treats all channels as exchangeable within a subject, so latent dimensions are not straightforwardly mapped back to hippocampal subdomains. This is also a problem with traditional hyperalingment and was addressed by Spotlight Hyperalignment (Guntupalli et al., 2016) mentioned in Section 2, which performs alignment on smaller subsets of features occupying similar regions across subjects. Adapting this methodology to our method could be a focus for future work.

Another limitation of **MuSCH** is that it treats all non-matching classes as equally dissimilar. In practice, some odors may be inherently more similar (e.g., adjacent in the sequence or sharing perceptual/neural features). Enforcing uniform separation can over-penalize "near" classes and distort the underlying class geometry. In future work this can be addressed by modifying the SupCon loss to downweight negatives from classes similar to that of the anchor. The following alter form of the SupCon loss could achieve this:

$$\mathcal{L}(i,j) = -\frac{1}{|P(i,j)|} \sum_{p \in P(i,j)} \log \frac{\exp\left(\hat{z}_{ij} \cdot \hat{z}_p / \tau\right)}{\sum_{a \in A(i,j)} \alpha_{ija} \exp\left(\hat{z}_{ij} \cdot \hat{z}_a / \tau\right)}, \tag{3}$$

with $\alpha_{ija} = 1$ for unrelated classes and $\alpha_{ija} \approx 0$ for very similar classes. Determining the similarity between classes could provide additional nuance to our method. Such a weighted SupCon loss could also address a limitation of **MuSCH**: that is can only be applied to datasets with a discrete class label. If $\alpha_{ija}$ were a function of a more complicated, continuous response.

**Future Directions.** Finally, our current pipeline collapses each trial's spiking activity in the 250–500 ms window into a single rate vector, discarding within-window timing differences (e.g., "early" vs. "late" responders). As a result, each odor is represented by a single point in latent space. Prior work shows meaningful temporal structure within this window (Shahbaba et al., 2022). A more faithful approach would model each trial as a short trajectory and align sequences rather than points. Future work in this direction relates to the class-weighted form of the SupCon loss function above because corresponding points along different subject's latent trajectories could be viewed as coming from the same class while points further apart in the trajectory could be made to be dissimilar with progressively larger values of $\alpha$.

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

DISCLOSURE OF THE USE OF LARGE LANGUAGE MODELS

The authors disclose the judicious use of Large Language Models to polish the writing and grammar of this paper.

REPRODUCABILITY STATEMENT

We recognize the importance of reproducibility when presenting our research findings. To support those ends, we have used a dataset that is publicly available online and shared its location. We also include the code we used as supplementary material. This includes the code for generating the simulated data as well as the code for fitting our proposed model and ablation models to both the simulated and real data. We document in the code what each function does.

# A  APPENDIX

## A.1  CONNECTION BETWEEN THE LOSS AND THE GENERALIZED HYPERALIGNMENT MODEL

Equation 1 should be understood as a modeling objective rather than a hard constraint: for each class $k$, we would like all subject-specific embeddings of the same class, i.e. $f_i(x_{ij})$ where $y_{ij} = k$, to concentrate around a shared, subject-invariant representation $S_k$, with residual variation captured by $\epsilon_{ij}$. The supervised contrastive loss in Equation 2 is designed so that any low-loss solution has precisely this structure: it pulls together all embeddings that share the same class label (regardless of subject) and pushes apart embeddings of different classes. If we define $S_k$ after training as the mean embedding of all examples with label $k$ across subjects, then at a low-loss solution we have $f_i(x_{ij}) \approx S_{y_{ij}}$ and $f_i(x_{ij}) - S_{y_{ij}}$ plays the role of $\epsilon_{ij}$. Thus **MuSCH** can be seen as implicitly instantiating the generalized hyperalignment model in Equation 1, with $S$ given by the class centroids and $f_i$ represented by subject-specific neural encoders trained under Equation 2. We do not claim a formal global convergence guarantee, which remains an open problem for deep contrastive objectives, but empirically we observe that the learned embeddings exhibit low within-class, cross-subject variance and high between-class separation (see Figure 8), consistent with Equation 1.

## A.2  TRAINING DETAILS FOR SUBJECT-SPECIFIC ENCODERS

For each subject, we train the encoder parameters using Keras/TensorFlow with the Adam optimizer to minimize the supervised contrastive loss in Equation 2. We use a learning rate of $\eta = 0.005$, batch size $B = 50$, and train for up to $E = 100$ epochs with early stopping. We stop if the training loss does not decrease by at least 0.3 for 10 consecutive epochs (patience = 10, min_delta = 0.3), and retain the encoder from the epoch with the lowest loss.

## A.3  SIMULATION RESULTS

Table 2 is the tabular form of the simulation results displayed graphically in Figure 3.

| | Percent non-encoding neurons ($p$) | | | | | |
|---|---|---|---|---|---|---|
| **Model** | 0% | 20% | 40% | 60% | 80% | 90% |
| **Raw-Single** | $0.962 \pm 0.094$ | $1.048 \pm 0.101$ | $1.129 \pm 0.090$ | $1.324 \pm 0.121$ | $1.696 \pm 0.109$ | $2.107 \pm 0.165$ |
| **Raw-Pooled** | $1.431 \pm 0.146$ | $1.655 \pm 0.161$ | $1.958 \pm 0.163$ | $2.475 \pm 0.238$ | $2.893 \pm 0.183$ | $3.015 \pm 0.232$ |
| **Aligned-Single** | $0.858 \pm 0.066$ | $0.929 \pm 0.077$ | $1.024 \pm 0.064$ | $1.226 \pm 0.085$ | $1.511 \pm 0.086$ | $1.761 \pm 0.088$ |
| **MuSCH** | $\mathbf{0.828 \pm 0.047}$ | $\mathbf{0.863 \pm 0.045}$ | $\mathbf{0.941 \pm 0.058}$ | $\mathbf{1.101 \pm 0.063}$ | $\mathbf{1.327 \pm 0.072}$ | $\mathbf{1.491 \pm 0.076}$ |

The lowest value for each $p$ is in **bold**.

Table 2: Categorical cross-entropy (mean $\pm$ SD) on the target subject's held-out test set across 60 random splits. Columns indicate the fraction of non-encoding neurons ($p$). Lower is better. Classification performance degrades as the signal contained in individual neurons is systematically reduced. However, **MuSCH** counteracts this effect by sharing information across subjects.

## A.4 APPLIED RESULTS

**A Note on Decoder Architecture and Linear Separability**. Our main experiments deliberately use the same compact nonlinear decoder (single hidden layer with sigmoid activation and softmax output) for all input types (**Raw-Single**, **Raw-Pooled**, **Aligned-Single**, **MuSCH**, **MMVM-VAE**), so differences in categorical cross-entropy reflect differences in the input representation rather than in decoder capacity (Sec. 4.3). To directly probe linear separability, we performed an ablation in which we replaced this network with a purely linear decoder (multinomial logistic regression) trained on the target subject's raw data (**Raw-Single**), all subject's pooled raw data (**Raw-Pooled**), the target subject's aligned embeddings (**Aligned-Single**) and all the subjects aligned embeddings (**MuSCH**). As shown in Figure 4, linear decoding achieves comparable categorical cross-entropy from the **Raw-Single**, **Aligned-Single** and **MuSCH** inputs. This indicates that the **MuSCH** embeddings do not produce a large gain in strict linear separability. Instead, **MuSCH** reorganizes the latent space in a way that is most beneficial to a shallow *nonlinear* decoder, which yields the performance improvements displayed in Figure 3, Figure 5, Table 2 and Table 1.

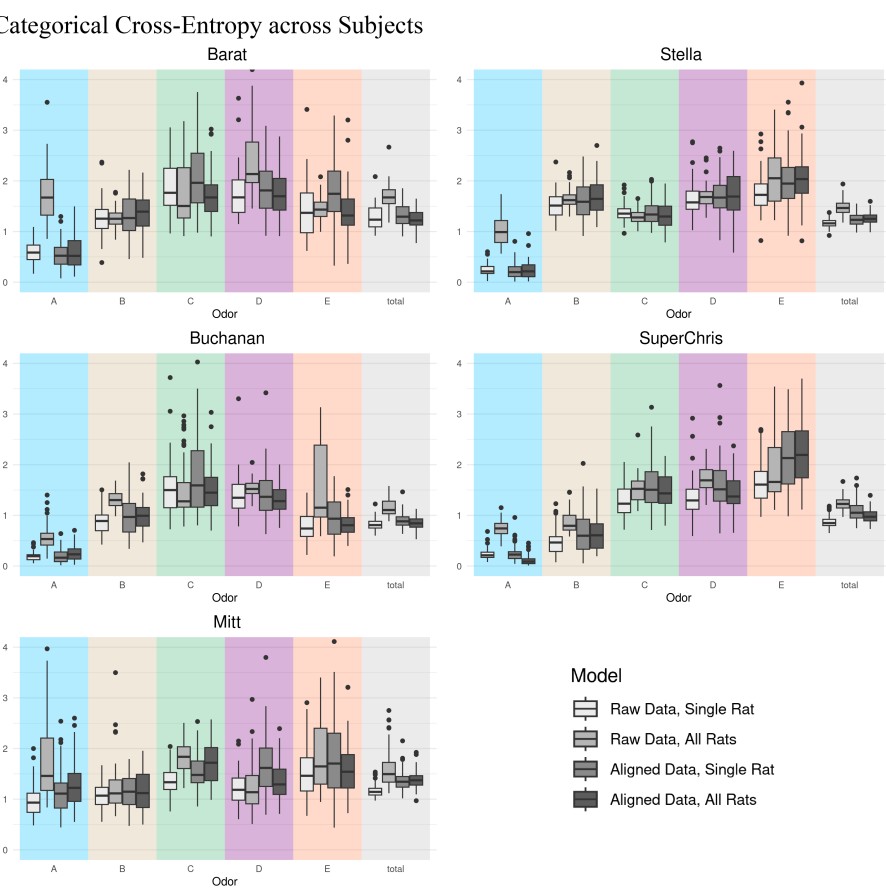

Figure 4: Decoding odor using a linear decoding model (multinomial logistic regression). There is no discernible improvement in categorical cross-entropy when the decoder is trained on multi-subject aligned latent embeddings rather than the raw data of the target subject. This indicates that MuSCH's main benefits are not due to increased linear separability but to a representation that is more effectively exploited by a shallow *nonlinear* decoder.

**Graphical display of applied results.** Figure 5 displays the distribution of categorical cross-entropy across the different rats, odors and models that was summarized in Table 1. Figure 6 displays these same results but without **Raw-Single** and **Raw-Pooled**, the presence of which detracts from our argument that SupCon multi-subject alignment improves the efficacy of latent embeddings.

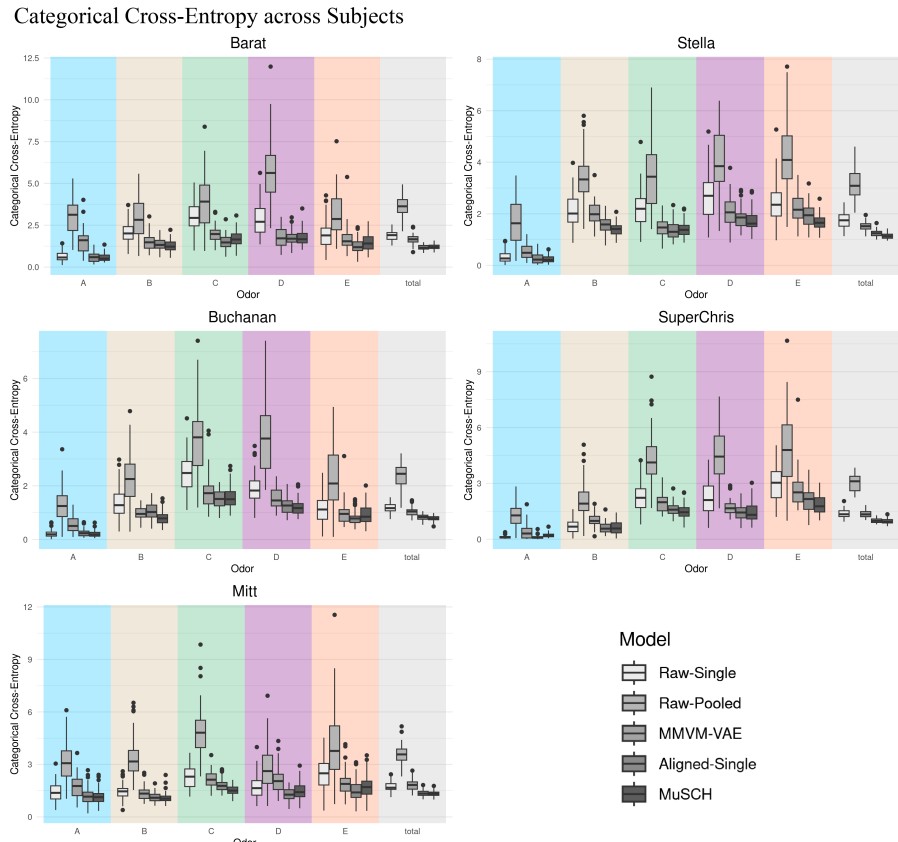

Figure 5: Applied Results. Comparison of downstream decoding performance as measured by categorical cross-entropy (cce) across the four methods described in Section 4 and also MMVM-VAE. The boxplots display the cce over 60 random test subsets for each subject (rat). The cce is displayed by odor class as well as overall (total).

## A.5 ADDITIONAL ANALYSIS ON LATENT EMBEDDINGS

We include additional analysis of the latent embeddings generated by our proposed **MuSCH** model. The SupCon loss function optimizes on cosine similarity, encouraging embeddings from the same class to be more similar as compared to similarity between embeddings from different classes. Cosine similarity values range from 1, maximum similarity, to -1, maximum dissimilarity. Figure 7 displays the distribution of cosine similarity between pairs of embeddings resulting from **MuSCH** applied to the real-world rat spike data. It shows that indeed pairs of embeddings from the same class tend to be closer to 1 than pairs from different classes. However, this figure also shows that the full range of cosine similarity is not being utilized. Different-class embedding pairs still tend to have high similarity, just not as much as same-class pairs. It would be better for downstream decoding if different-class pairs were dissimilar, i.e. with negative cosine similarity. That we don't see that could reflect a limitation of the simple encoding networks employed. It could also be characteristic of this dataset, that the neural signal for different odors are inherently similar.

Figure 8 displays the PCA projections of both a target subject's raw data and the latent embeddings generated by **MuSCH**. This figure demonstrates the improvements in class separability that make downstream decoding easier.

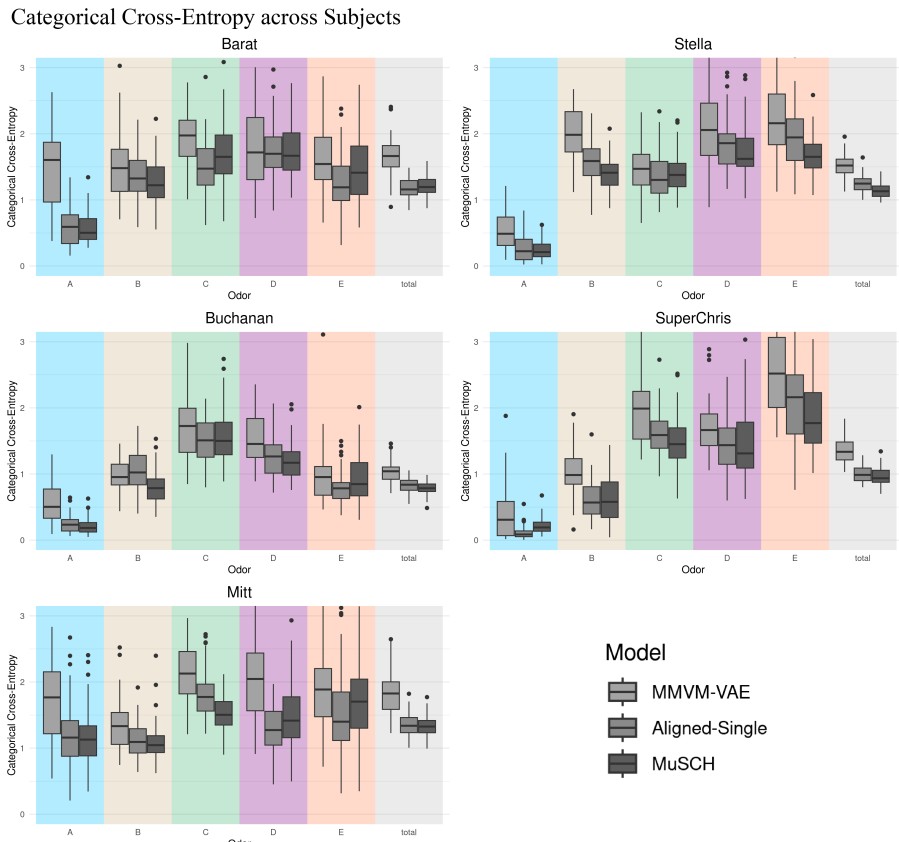

Figure 6: Applied Results. Comparison of downstream decoding performance as measured by categorical cross-entropy (cce) focusing on just MMVM-VAE, **Aligned-Single** and **MuSCH**. This plot demonstrates the general improvement of **Aligned-Single** over MMVM-VAE and the further improvement of **MuSCH** in many situations.

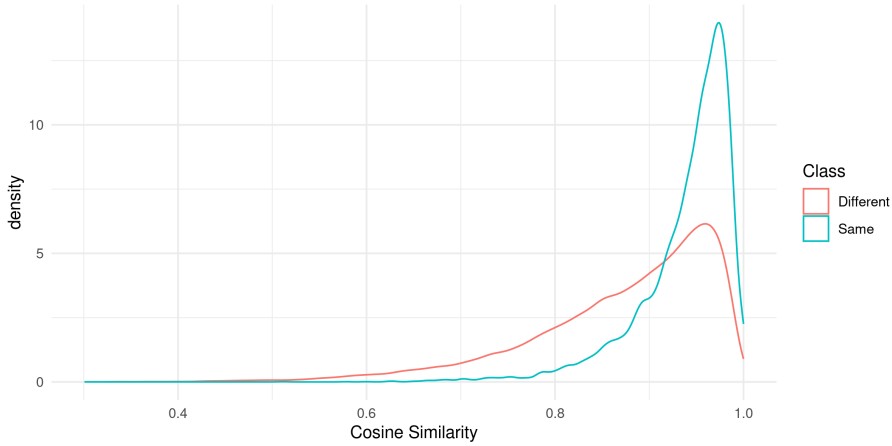

Figure 7: **Distribution of Cosine Similarity** Pairs of samples from the held-out testing subset are passed through the trained subject-specific encoding networks. The cosine similarity of the resulting latent representations are calculated. Sample pairs coming from the same class tend to have a higher similarity than samples pairs coming from different odor classes. This illustrates the same-class clustering behavior that is difficult to visualize directly because of the 15-dimensional embedding space.

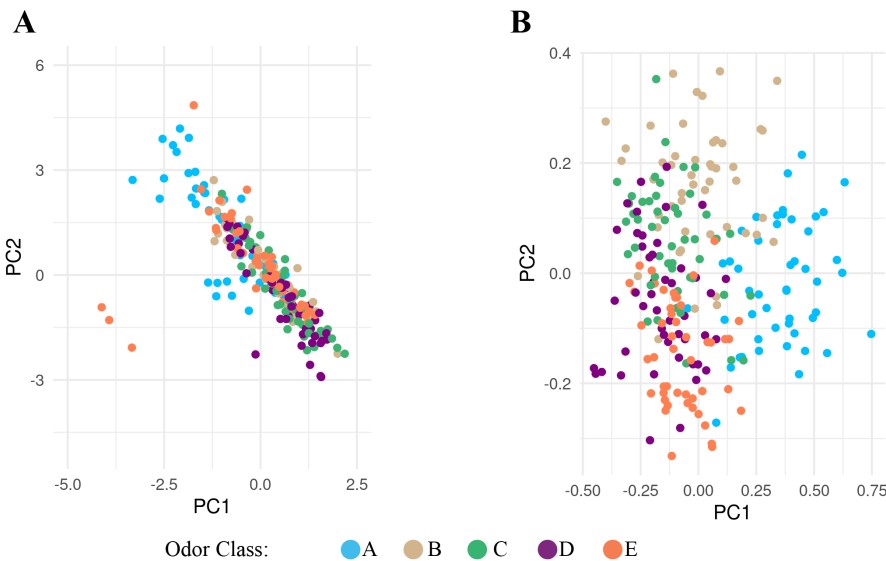

Figure 8: Clustering behavior of latent embeddings. (A) PCA projection of a single rat's raw data, before any efforts at alignment. (B) PCA projection of the latent embeddings of that same rat's data after applying MuSCH. The embeddings for trials from the same odor class show clustering behavior which explains the increased performance of downstream decoding.