# OpenReview forum: "Neural Decoding through Multi-subject Class-conditional Hyperalignment"
_ICLR.cc/2026/Conference — Submitted to ICLR 2026_

### Official Review · Reviewer_GJQJ · 2025-10-20

**Soundness:** 2
**Presentation:** 4
**Contribution:** 1
**Rating:** 2
**Confidence:** 5

**Summary:**

The authors sued supervised contrastive learning and by formulating positive pairs from data across different subjects that share classes. Through this loss objective, a separate encoder was trained for each subject to hyperalign the latent space of neural data across subjects. The proposed method is tested in simulated data and rats' hippocampal spike data performing odor memory task.

**Strengths:**

* The problem is well defined and the limitation of the previous methods are well described.

* The work demonstrates successful implementation and training of supervised contrastive learning that uses across-subject pairs to encourage hyperalingment.

**Weaknesses:**

* The definition of equation 1 is not used or proven. It is unclear that the mathematical ground or any empirical evidence that the loss in equation 2 will guarantee (or likely converge to) that the solution will converge to the right hand side of equation 1. As an arbitrary neural network can express an arbitrary function, grounding evidence is crucial that demonstrates the hyperalignment definition in eq 1 can be achieved.

* Mixed results in ablation experiment: putting aside the groundness of the proposed method, the decoding results denoted in Table 1 show that including the proposed training method yields degraded performance from the w/o proposed method (“Aligned-Single”), for many cases. This may imply the proposed loss is not very well compatible or suboptimal in hyperalignment purpose.

* The model evaluation is significantly limited in terms of recruiting baseline models. Only MMVM-VAE is tested. There are other hyperalignment methods listed in the related works. (Some of them need timely aligned stimuli but the dataset can be easily processed to provide such by chunking time span of each category, and dynamic time warping can be used.) Moreover, how the authors trained the model using the dataset they are testing is not elaborated. If they used a pretrained model, it should be made sure that the training/testing distribution exactly matches to provide a fair comparison.

* The proposed methods still require class information and cannot be applied to complex stimuli or behaviors which are hard to define discrete categories.This significantly limits the utility of the method.

* The approach lack novelty that it simply reorganize positive/negative pairs to be recruited across subjects. Due to the above caveats in lack of ground and limited evaluation, this lack of technical novelty is yet to be resolved.

**Questions:**

See weaknesses.

---

> ### Author Response · Authors · 2025-11-25
> **We clarify various points in the manuscript, address mixed results in ablation study, provide justification for the use of SupCon loss for multi-subject alignment, and advocate for the novelty of our method.**
>
> We thank the reviewer for their careful reading. Our responses are below.
>
> **Equation 1: role and usage** Equation 1 is not a theorem but a generalized hyperalignment model that we use as a unifying framework for classical hyperalignment, SRM, and MuSCH. In the revision, we explicitly name it as such, explain how particular choices of $f_i$ and $S$ recover classical hyperalignment and SRM, and state that MuSCH is another instance of this framework with subject-specific neural encoders and a supervised contrastive objective.
>
> **Justification of SupCon loss** We agree this was underexplained and have added a justification in the Appendix. Briefly, Eq. 1 describes the desired structure: for each class $k$, embeddings $f_i(x_{ij})$ with $y_{ij}=k$ concentrate around a shared representation $S_k$, with residual variation $\epsilon_{ij}$. The supervised contrastive loss in Eq. 2 is designed to encourage exactly this, by pulling together same-label trials across subjects and pushing apart different labels. We do not claim a formal convergence guarantee, but this explains why Eq. 2 is a natural objective for instantiating the generalized hyperalignment model.
>
> **Mixed results in ablation experiment** Table 1 shows mixed effects when comparing MuSCH to the single-subject baseline (“Aligned-Single”): in some rat–odor combinations MuSCH improves accuracy, while in others the single-subject model is slightly better. We view this as expected rather than evidence that the loss is incompatible with hyperalignment. Rats exhibit substantial inter-subject variability in tuning, trial history, and noise. When cross-subject structure is consistent, sharing information via MuSCH helps; when it is not, the shared prototype can dilute subject-specific features and lead to modest drops. Thus we do not expect multi-subject alignment to uniformly dominate the single-subject baseline. We clarify this interpretation in Section 5 and emphasize that heterogeneous effects across rats are informative about subject specificity rather than a failure of the objective.
>
> **Additional baselines** We agree that more baselines are useful. We already include MMVM-VAE, a state-of-the-art variational multi-view model, trained from scratch on our dataset using the same train/test splits as MuSCH (no external pretraining), ensuring a fair comparison. Following reviewers’ suggestions, we now also discuss transformer-based multi-subject alignment models such as POYO in the revised manuscript.
>
> Classical hyperalignment assume a continuous, time-locked stimulus shared across subjects. In our dataset, odor presentation is self-paced and randomized; stimulus onsets and numbers of repetitions differ across rats. Constructing a pseudo time-locked stimulus via chunking and dynamic time warping would require discarding many trials or imposing arbitrary trial matchings, effectively fabricating a shared stimulus trajectory and pushing these methods outside their intended regime. MuSCH is explicitly designed for this label-based, non–time-locked setting, aligning subjects using class labels without requiring trial- or time-wise correspondence. We clarify this distinction and expand our MMVM-VAE training description in the revision.
>
> **Labeled data requirement and novelty** We agree that MuSCH requires discrete class labels. However, by removing the need for a separate time-locked training dataset (required by classical hyperalignment), we expand the range of datasets that can be aligned. Many neuroscience studies are organized around discrete conditions (stimulus identity, choice, reward, rule, etc.), where labels are naturally available. Our formulation is not suited to fully continuous, unlabeled behavior; we now highlight this as a limitation and briefly discuss possible extensions (e.g., modified contrastive objectives or weaker forms of supervision) in the Limitations section.
>
> MuSCH deliberately builds on the standard supervised contrastive loss and does not introduce a new objective function. The contribution lies in (i) the generalized hyperalignment model (Eq. 1), which unifies classical hyperalignment and SRM as special cases with linear mappings and time-index “labels”, and (ii) a new instantiation of this framework in a different regime. In MuSCH, subject-specific neural encoders $f_i$ are learned jointly using a SupCon objective that treats same-condition trials across subjects as positives, aligning them to class-level representations without requiring a continuous, time-locked stimulus. While at the loss level MuSCH “reorganizes” positive/negative pairs, this is precisely what turns a single-subject SupCon encoder into a multi-subject alignment mechanism consistent with Eq. 1. Many controlled experiments supply exactly this kind of discrete condition information but lack a shared continuous stimulus, so our label-based, non–time-locked formulation fills a gap not addressed by existing hyperalignment methods.

---

### Official Review · Reviewer_ETex · 2025-10-27

**Soundness:** 2
**Presentation:** 3
**Contribution:** 2
**Rating:** 4
**Confidence:** 5

**Summary:**

The study introduces a supervised contrastive learning framework for class-conditional functional alignment across subjects that does not have the secondary dataset requirement in hyperalignment method. The suggested method jointly class-conditionally-clusters and encodes samples in a lower-dimensional space with supervised contrastive learning, prior to classification.

**Strengths:**

- The paper proposes a novel supervised contrastive method for hyperalignment problem.
- The application domain is novel, introducing new challenges for the field.

**Weaknesses:**

- Regarding the optimization of the subject-specific encoders, the stopping criterion is not explained in the study. The optimization of encoders and their utility is data dependent, hence a metric is needed to quantify convergence for reproducibility.
- Since the evaluation is dependent on the chosen decoder architecture, an ablation of network parameters in Section 5 is missing and required for isolating alignment performance from decoder capacity.
- The decoder that is used to assess how well the suggested method works, is a nonlinear neural network, while the encoders are linear networks. Since a nonlinear neural network gradient is input-dependent, due to saturation mode of the sigmoidal function, its performance is not a reliable identifier of how well the data is separable. In an ablation study, adding the results of a linear decoder is required to more directly measure the improvement in linear separability.
- The positioning of the study could be strengthened by discussing closely related studies on supervised functional alignment implementations in Canonical Correlation Analysis (CCA) studies, specifically Maxvar-Generalized-CCA and variational-CCA implementations.
- An overall method figure for Section 3 is missing, making it harder to grasp the core contribution of the paper.
- T is introduced without a definition (Line 134-135) that stands for the number of frames in movie-watching task.

**Questions:**

Q: Could you please specify the convergence criteria used for optimizing the subject-specific encoders? For example, was it based on early stopping using a validation loss, a fixed number of epochs, or a performance threshold? Was the target subject held-out data used in the assessment of convergence? Clarifying this is essential for assessing the robustness of the training procedure and for ensuring the results are reproducible.

---

> ### Author Response · Authors · 2025-11-25
> **We address our optimization process, include an ablation of our decoder model, include a brief discussion of CCA methods, and provide clarification.**
>
> We thank the reviewer for the thoughtful consideration of our manuscript. We address the points you raised below.
>
> **Regarding the optimization of the subject-specific encoders...** We acknowledge the importance of including further details about our choice of optimization mechanism, especially early stopping criteria, which we did employ, learning rate, batch size, etc. for reproducibility. We have clarified the optimization and convergence criterion for the subject-specific encoders in the Appendix and now explicitly refer to this appendix in the Encoder Details section.
>
> **...an ablation of network parameters in Section 5 is missing...** We thank Reviewer 3 for raising this important point. In all main experiments in Section 5, we intentionally use the same compact nonlinear decoder (single hidden layer with sigmoid activation and a softmax output layer) for all input types (Raw-Single, Raw-Pooled, Aligned-Single, MuSCH, MMVM-VAE), so differences in categorical cross-entropy reflect differences in the learned representations rather than decoder capacity (Sec. 4.3). Following the reviewer’s suggestion, we have added an explicit ablation on decoder capacity by replacing this network with a purely linear decoder (multinomial logistic regression) trained on 1) the target subject’s raw features and 2) the MuSCH multi-subject aligned embeddings. The new results (Fig. 7 in the Appendix) show that linear decoding performance is comparable in both cases, indicating that MuSCH does not substantially improve strict linear separability. Instead, MuSCH organizes the latent space so that a shallow nonlinear decoder can better exploit class structure, which is where we observe the largest gains. We include details in the Appendix to make this ablation explicit and clarify the respective roles of representation alignment and decoder architecture in Sec. 4.3.
>
> While we agree that sigmoids can in principle suffer from saturation, in our shallow non-linear decoder we did not observe unstable or highly variable training. Across 60 random train/test splits, the categorical cross-entropy shows low variance, indicating that optimization is stable and not overly sensitive to initialization or split. Together with our linear decoder ablation (Fig. 7), this suggests that the gains we report are not driven by pathological saturation effects but by genuine structural differences in the learned representations.
>
> **The positioning of the study could be strengthened by discussing closely related studies...** We appreciate the suggestion to discuss supervised functional alignment methods based on (MaxVar-)GCCA and variational CCA. These approaches also learn subject-specific projections into a shared latent space, and stimulus-informed GCCA (SI-GCCA) in particular incorporates a common stimulus representation as an additional view. However, such methods fundamentally assume that all views share the same set of instances (all views correspond to a common continuous stimulus), so that samples can be indexed identically across subjects. In our experiments, each subject performs a self-paced odor-identification task. The odors are typically presented in the same order (ABCDE) but not always and the rat initiates the presentation of the next odor. Therefor, there is no natural one-to-one correspondence between trials (or time points) across subjects; only the class labels are shared. Applying (SI-)GCCA would therefore require additional aggregation or matching steps (such as collapsing to per-condition averages), which would change the problem by discarding trial-level variability. MuSCH is explicitly designed to handle this unaligned, label-based setting by using supervised contrastive learning to align subjects via class labels rather than via shared stimulus time courses. We clarify this distinction and add a short discussion of (MaxVar-)GCCA and variational CCA in the introduction.
>
> **An overall method figure for Section 3 is missing, making it harder to grasp the core contribution of the paper.** We agree that an overview illustration is important for understanding the core contribution. Our revised manuscript now makes explicit that Fig. 1A serves as the overall method figure for MuSCH. It depicts the subject-specific encoders, the supervised contrastive alignment across subjects, and the decoder trained on the aligned embeddings of the target subject. We have updated the caption of Fig. 1 and added a pointer after introducing our model in Section 3 (“An overview of the MuSCH framework is shown in Fig. 1A”) to make this clearer.
>
> **T is introduced without a definition (Line 134-135)...** We appreciate Reviewer 3 raising this point. What we meant by $T$ is the total number of measurements taken from all subjects as they were exposed to the same time-locked stimulus (watching a movie). We have clarified this in the manuscript.

---

### Official Review · Reviewer_pXka · 2025-10-31

**Soundness:** 2
**Presentation:** 2
**Contribution:** 2
**Rating:** 4
**Confidence:** 3

**Summary:**

This paper introduces MuSCH (Multi-Subject Class-conditional Hyperalignment), a method for aligning neural data across multiple subjects using contrastive learning. This eliminates the need for secondary alignment datasets required by traditional hyperalignment methods by leveraging class labels (supervised) from the primary experimental dataset.

**Strengths:**

- The paper tackles a limitation of existing hyperalignment methods—the requirement for expensive, time-consuming secondary datasets with time-locked stimuli. This is especially problematic for animal studies where data collection is hard

- Reproducibility: The author used public dataset and shared with code.

**Weaknesses:**

1. Supervised contrastive learning is a good way to train the model, however, recent works in self-supervised learning (foundation model) in neural data also show extremely well performance for cross-subject/animal/session results. Compare with one of state-of-the-art decoding methods in this field, e.g. POYO+ (ICLR2025), NEDS (ICML2025).

2. Scalability Concerns. What about scale up the method training with 10, 50 animals? Does the N different encoder network still works well?  Computation might be costly?

3. Only one real dataset result. I'm not sure about model's generalization ability.

**Questions:**

- What's the representation of neural data looks like? Are there any clustering effects for subjects or similar tasks?

- Can you show some further analysis on the representations?

---

> ### Author Response · Authors · 2025-11-25
> **We address recent multi-subject neural decoding models, add requested visualizations and analyses to the Appendix, and explain our use of a single dataset.**
>
> We appreciate your thoughtful consideration of our manuscript. Thank you for alerting us to some recent transformer-based multi-subject neural decoding models (POYO, POYO+, NEDS). Reviewer 1 (above) also mentioned these and we discussed similarities to our method there.
>
> **What's the representation of neural data looks like? Are there any clustering effects for subjects or similar tasks?**
> There are clustering effects of latent embeddings of trials having the same odor class. This is induced by the Supervised Contrastive Loss function, which pulls together embeddings of the same odor class, regardless of which subject the neural response came from, and pushes apart embeddings from different classes. We agree that a visualization of these embeddings is helpful to see the effect of the SupCon loss function and have included a visualization in the Appendix. We found PCA projections of both a single subject's raw data and the latent embeddings after multi-subject alignment. The degree of separation between the classes is less pronounced because the first two principal components focus on the contrast between odor A and the other odors. This signal is strongest in the data for reasons explained in the paper. Separation of the other odors happens in higher dimensions which are not visible but which the decoder model has access to and utilizes.
>
> **Scalability Concerns. What about scale up the method training with 10, 50 animals? Does the N different encoder network still works well? Computation might be costly?**
> We agree that it is important to consider how MuSCH scales with the number of subjects. In most systems and cognitive neuroscience datasets, the number of animals or participants per experiment is relatively small (often fewer than 10–15), and even recent multi-subject alignment methods such as POYO and its extensions are evaluated on datasets with only a handful of subjects. In this situation, having one encoder per subject is computationally manageable.
>
> Concretely, all subject-specific encoders in MuSCH share the same architecture (number of layers and hidden units). If a single encoder has $P$ parameters, the total encoder parameter count grows linearly with the number of subjects $N$, i.e., $\mathcal{O}(NP)$. Training cost similarly scales approximately linearly in $N$: as more animals are added, the number of training examples increases and each example incurs a fixed-cost forward/backward pass through the corresponding encoder. These computations could be parallelized across subjects on modern GPUs, so wall-clock time remains practical for $N$ in the tens. We coded our model to run on a CPU and it takes only several minutes to run for 5 subjects with a total of 953 total samples. We do not feel the number of subjects will lead to computational constraints in any typical neuroscientific dataset.
>
> **Only one real dataset result. I'm not sure about model's generalization ability.**
> We agree that evaluating MuSCH on additional real datasets would further strengthen the empirical evidence. In this work, we chose to focus on a single but challenging hippocampal spike dataset spanning multiple rats and sessions, complemented by a controlled simulation study where we systematically varied cross-subject structure and could directly inspect the learned representations. Together, these experiments allow us to test MuSCH both in a realistic neural recording setting and in scenarios where ground truth alignment structure is known.
>
> While there are many publicly available multi-subject electrophysiology datasets that match our setting (label-based, non–time-locked trial structure with multiple conditions), such as on CRCNS.org. However, applying our model to a new dataset requires a non-trivial amount of preprocessing. We therefore view applying MuSCH to additional datasets as an important but separate piece of work, and we will explicitly highlight this as a limitation and direction for future research in the revised manuscript. Our goal in this paper is to introduce the framework, connect it to existing hyperalignment methods, and provide an initial validation on simulations and a realistic multi-rat dataset, rather than to exhaustively benchmark across many different experiments.
>
> **Can you show some further analysis on the representations?**
> We have added a PCA plot of latent representations in the Appendix, in response to a question from Reviewer 3. We have also included, also in the Appendix, an analysis and plot of the distribution of cosine distance between latent representations from the same class versus different classes. Cosine distance is the measure of similarity used by the SupCon loss function. The difference in distributions illustrate the clustering behavior that is difficult to visualize directly because the latent dimension we chose to be 15. The cosine similarity tends to be higher for sample pairs coming from the same class.

---

> > ### Comment · Reviewer_pXka · 2025-11-27
> >
> > I want to thank the authors for the thoughtful response and for clarifying the latent-space visualization. Given the dataset limitation, I understand it's fine to conduct analysis on current real dataset.
> >
> > However, as noted above, the lack of sufficient comparisons for MuSCH leaves me uncertain about the method’s true capabilities. Without benchmarking against existing methods, it remains difficult to assess the extent to which MuSCH offers an advantage.
> >
> > Additionally, I want to clarify techniques such as subject embeddings and neuron embedding, which are designed for cross-subject and various neuron numbers, typically adding only 𝑂(𝑃) training cost. This could be a future direction to explore.

---

### Official Review · Reviewer_GQnF · 2025-10-31

**Soundness:** 2
**Presentation:** 2
**Contribution:** 2
**Rating:** 4
**Confidence:** 3

**Summary:**

The paper introduces MuSCH (Multi-Subject Class-Conditional Hyperalignment), a method for aligning neural data across subjects using Supervised Contrastive Learning based on class labels instead of time-synchronized stimuli. This approach eliminates the need for secondary alignment datasets and enables cross-subject neural decoding in nonhuman studies. The method is evaluated on both simulated datasets and real hippocampal spike data from rats performing an odor sequence-memory task. Results show that MuSCH improves downstream decoding accuracy.

**Strengths:**

Strengths:
1. The use of Supervised Contrastive Learning (SupCon) for multi-subject alignment is new in neuroscience applications.
2. The method is clearly formulated and mathematically consistent.

**Weaknesses:**

Weaknesses:
1. The problem of aligning neural representations across subjects is not new, as several existing methods, such as [1][2][3], already address or could feasibly address multi-subject representation learning under different assumptions. However, the authors do not sufficiently discuss these related approaches or clearly delineate how their method conceptually differs from them.
2. The authors do not compare MuSCH with closely related multi-subject or multi-session decoding models.



References:
[1] Azabou, Mehdi, et al. "A unified, scalable framework for neural population decoding." Advances in Neural Information Processing Systems 36 (2023): 44937-44956.
[2] Zhang, Yizi, et al. "Towards a" universal translator" for neural dynamics at single-cell, single-spike resolution." Advances in Neural Information Processing Systems 37 (2024): 80495-80521.
[3] Zhang, Yizi, et al. "Neural encoding and decoding at scale." arXiv preprint arXiv:2504.08201 (2025).

**Questions:**

Questions:
1. How are positive pairs and negative pairs exactly sampled within a batch?
2. Why the encoders are frozen before training the decoder, rather than training the whole model end to end? It’s unclear how this design choice affects the adaptability of the learned representations and overall decoding performance.

---

> ### Author Response · Authors · 2025-11-25
> **We address recent multi-subject decoding methods, the composition of our training batches, and our choice to train the encoders and decoder separately.**
>
> We appreciate your thoughtful consideration of our manuscript. We have addressed the three points raised in your review.
>
> **How are positive pairs and negative pairs exactly sampled within a batch?**
> The composition of a training batch is an important consideration with self-supervised contrastive loss approaches because suitable negative samples must be found in the training data to contrast against the positive pair (a sample and an augmentation of that same sample). This search requires care because there are no explicit labels to help identify negatives. However, in Supervised Contrastive Learning (our situation) we do have explicit labels (the odor presented to the rat for that particular trial). Forming a batch by randomly sampling from all available trials across all subjects guarantees that there will be multiple positives (there are only 5 distinct odors and our batches contain 50 trials). It is also virtually impossible to have no negatives (this can only happen if all 50 samples were the same odor). Therefor, a random sampling mechanism is sufficient to form training batches.
>
> **Why the encoders are frozen before training the decoder, rather than training the whole model end to end? It’s unclear how this design choice affects the adaptability of the learned representations and overall decoding performance.**
> The encoders are frozen before training the decoder for several reasons. Our method addresses the problem of non-exchangeablility of trial firing rates across subjects, preventing any joint analysis of multi-subject neural data. We were less focused on what the downstream joint analysis could be and wanted a general-purpose embedding that could be treated as input to any number of downstream analyses. The choice to freeze the encoder was also done in the original Supervised Contrastive Learning paper for similar reasons. However, our choice of downstream task is decoding the same stimulus that the SupCon loss function was trained with. Therefore, the alignment stage and the decoding stage's objectives are concordant.
>
> **The authors do not sufficiently discuss these related approaches or clearly delineate how their method conceptually differs from them. The authors do not compare MuSCH with closely related multi-subject or multi-session decoding models.**
> We thank Reviewer 1 for bringing to our attention POYO, POYO+ and NEDS as state-of-the-art decoding models of multi-subject neural spike data. We were unaware of these methods as their aim is not explicitly to learn general latent representations of neural responses that are shared across subjects. Instead, these methods perform implicit alignment of individual neurons according to their computational role in a prespecified behavior. These methods can be adapted to decode experimental conditions (such as which odor was presented to a rat during a trial, in our case). However, an important difference between these methods and our own is their focus. POYO, POYO+ and NEDS focus on decoding behavior from neural spike data by learning patterns of activation that are shared across multiple subjects and studies. The focus of our method, however, is on learning shared representations of neural responses to stimuli so that any arbitrary joint analysis is possible. We use downstream decoding to demonstrate that information is being shared across subjects but the object of interest is in these learned latent representations. The transformer models mentioned yield predictions and not latent representations. We have included text to that effect in the manuscript. Another potential difference between these transformer-based methods and our own Supervised Contrastive Hyperalignment is the amount of data necessary for adequate training. The POYO paper uses 27,373 "units" and 16,473 neuron-hours for training while our dataset has only 41 "units" and 31 neuron-minutes. We feel the scale of our data is typical of neuroscientific studies. Foundational models for decoding neural spike data require much more data than what is currently available.

---

> ### Comment · Reviewer_pXka · 2025-11-27
>
> Thank you to the authors for the clear responses. I now understand the batch sampling procedure. However, I still believe the paper does not provide a fully fair comparison with existing methods, and several claims remain insufficiently supported.
>
> 1. On the stated goals of neural foundation models:
> The authors claim that prior neural foundation models aim “is not explicitly to learn general latent representations of neural responses that are shared across subjects.” This is not accurate. A core purpose of these models is to overcome variability across animals and recording sessions. Many of them incorporate subject/session embeddings or similar mechanisms designed to produce more consistent latent representations across subjects.
>
> 2. On the contribution regarding shared information:
> The manuscript states that the main contribution is measuring shared information across sessions/subjects. Could the authors provide additional empirical analysis or theoretical justification demonstrating that MuSCH yields a superior latent geometry compared to existing approaches? Without such evidence, the claim remains invalid.
>
> 3. On comparisons with existing models:
> It is not necessary to fully train a neural foundation model to enable comparison. One could train the same architectures from scratch under the MuSCH framework, or start from available pretrained weights and fine-tune them. Therefore, it is unclear why comparisons with other methods are omitted. If MuSCH underperforms other models in decoding tasks, then stronger neuroscientific explanations or evidence (as mentioned in point 2) would be required to justify its advantages.

---

> ### Comment · Reviewer_GQnF · 2025-11-27
>
> I thank the authors for the detailed responses. My concerns regarding batch sampling and the frozen encoders have been resolved. However, my concern about the fairness of the baseline comparisons remains. I agree with Reviewer pXka that fully training neural foundation models is not necessary for comparison since one could train the same architectures from scratch within the MuSCH framework to assess their representational/decoding ability. In the absence of such direct and controlled comparisons, and without stronger evidence that MuSCH gives superior latent representations, I will maintain my current score.

---

### Author Response · Authors · 2025-12-02
**Summary of Discussion and Changes**

We thank the reviewers for their thoughtful, constructive, and encouraging feedback and for engaging carefully with the manuscript. We have revised the manuscript to directly address the major concerns raised and to improve clarity, reproducibility, and interpretability. We are grateful for the opportunity to clarify the scope and contributions of MuSCH and to strengthen the empirical and analytical support for our claims.

**Baseline and fairness.** We compare against MMVM-VAE, which is the most closely related state-of-the-art baseline for cross-subject alignment of neural spiking embeddings. We clarify the baseline training procedure and ensure a fair comparison by training MMVM-VAE from scratch on the same dataset and using identical train/test splits (no external pretraining or additional data).

**Summary of revisions.** In the updated manuscript we:

* Clarify the relationship between Eq. (1) and prior hyperalignment/SRM formulations and explicitly position MuSCH as an instance of a generalized hyperalignment framework.

* Add a justification connecting the supervised contrastive objective (Eq. (2)) to the desired class-conditional shared structure described in Eq. (1).

* Add new latent-space analysis and visualization showing similarity structure and geometry of learned embeddings to support the alignment behavior beyond downstream decoding.

* Add missing training/optimization details (optimizer, stopping criterion/early stopping, and other implementation details) to improve reproducibility and address concerns about convergence.

We believe these revisions substantially strengthen the paper and address the key issues raised in review. We appreciate the Area Chair’s consideration.

---

### Meta-Review · Area_Chair_VKtP · 2026-01-05

**Summary:**

This manuscript presents a multi-encoder neural network architecture aimed at learning a single shared latent space across multiple individuals. The model is trained with a contrastive loss to produce universal classification results. The authors test their method on simulated data, followed by an application of their model to neural recordings of rats performing an odor task. They compare their to a few baselines, however as the reviewers noted the full set of comparisons is limited with respect to the current literature on multi-subject "foundation model" type models. This stands out as one of the main weaknesses noted by multiple reviewers before and after the rebuttal (until the discussion was cut short). This was the main (although some of the other aspects could use more work) point that makes me feel that additional work is required for this manuscript to be accepted at ICLR.

**Reviewer Concerns:**

There were a number of concerns raised by the reviewers, including:
 1) Clarity of some of the assumptions, model choices, and definitions
 2) Missing references to very related work
 3) Missing comparisons to prior art in this area.
 4) Lack of sufficient ablation experiments

The authors did put effort in clarifying some of the points in the manuscript that the reviewers found unclear or confusing. The authors also added ablation experiments and discussions to address these points. The most critical point, however, was the comparisons with existing methods which remains outstanding.

**Reviewer Scores:**

The scores for this paper were 4,2,4,4. Given that two of the reviewers (4,4) declined to increase their score, I would assume that the remaining reviewers that did not yet respond would react similarly and retain the 4,4,2,4 spread.

---

### Decision · Program_Chairs · 2026-01-26

Reject